# Small-Sided Games as a Methodological Resource for Team Sports Teaching: A Systematic Review

**DOI:** 10.3390/ijerph17061884

**Published:** 2020-03-13

**Authors:** Carlos Fernández-Espínola, Manuel Tomás Abad Robles, Francisco Javier Giménez Fuentes-Guerra

**Affiliations:** Faculty of Education, Psychology and Sport Sciences, University of Huelva, 21071 Huelva, Spain; carlos.fernandez@ddi.uhu.es (C.F.-E.); jfuentes@uhu.es (F.J.G.F.-G.)

**Keywords:** teaching for understanding, team sports, technique, tactical behaviour

## Abstract

New models for teaching sports have arisen in the last years, characterised by the use of more contextualised situations, modified games, tactical awareness, transference of technical–tactical learning and different teaching progression, among other aspects. In this regard, small-sided games must be highlighted, due to their ability to integrate physical fitness, technique and tactical behaviour stimuli in similar conditions to the real game. Therefore, the aim of this systematic review was to analyse and describe the methodological possibilities that SSGs can provide regarding the teaching of technical–tactical aspects in team sports at young ages. The guidelines of the PRISMA declaration were followed with the purpose of conducting a systematic search. The search was performed in the databases *Pubmed*, *Web of Science*, *Scopus* and *SportDiscus*. From the 451 identified in an early phase, plus the 20 found in the references of other studies, only 47 met the inclusion criteria and were selected. The results yielded scientific evidence that justifies the use of small-sided games as a methodological resource for sports teaching at young ages. Among the main reasons, it can be highlighted that a reduction in the number of players and in the size of the pitch area increases the total ball contact per player and, therefore, the number of technical actions. Moreover, the intentional modification of certain rules helps to develop some sport training fundamentals.

## 1. Introduction

The evolution from traditional sport models to alternative approaches is well-founded [1]. These new proposals focus on various methodological aspects, such as the use of modified games [2], the integration of skills in contextualised situations [3], the transference of technical–tactical learning between similar sport modalities [4], from tactics teaching to the teaching of technique [5], the learner’s cognitive involvement [3], learning progression [6], and problem resolution [7,8]. From the aforementioned elements, the use of small-sided situations is one of the most important, given that this type of teaching allows coaches of all levels to teach technical and tactical skills, as well as to achieve an improvement in physical capacities [9,10]. 

Clemente, Martins and Mendes [11], Hill-Hass et al. [10], and Sgrò et al. [12], defined small-sided games (SSGs) as playful situations involving movement that are used for sport teaching/training, involve a smaller number of players per team, and are played on reduced spaces and with rules modified on purpose depending on the goals to be achieved, but respecting the main game principles. The difference between small-sided games and the traditional approach resides in executing skills as they appear in real competition [13,14].

Several systematic reviews have analysed the effects of modifying the above-mentioned variables or others like player’s experience [15] on technical–tactical aspects [12,16] or physical fitness [10]. Based on the scientific literature, and mostly on that involving adults, it can be concluded that several physical responses in football have been achieved through the use of SSGs, such as changes in heart rate, lactate concentration or perceived exertion [17,18,19]. Furthermore, other studies have reported changes in technical aspects like the number of passes, dribbles, shots or interceptions [20,21]. SSGs appear to be an effective strategy for training technical and tactical skills in young players of team sports [22], and “the manipulation of task constraints seems to be an effective strategy for creating practice environments that facilitate the acquisition of tactical principles” [16] (p. 13). However, the requirements (technical, tactical, etc.) of an SSG are different depending on the age, level of the players [23] and sport. In line with this, more research is required [24].

Therefore, the aim of the present study was to conduct a systematic review to analyse and describe the methodological possibilities that SSGs may offer regarding the teaching of technical–tactical aspects to young athletes in team sports.

## 2. Materials and Methods 

This systematic review on the methodological possibilities that SSGs may offer regarding the teaching of technical–tactical aspects to young athletes in team sports has been conducted following the PRISMA declaration and its practical guide to systematic reviews with or without meta-analysis [25,26].

### 2.1. Eligibility Criteria

The inclusion criteria applied for study selection were: (a) SSGs used as a methodological resource related to technique and tactics; (b) participants of a young age U-18, in sport development phase (sports initiation); (c) papers written in English or Spanish. In accordance with these criteria, papers found through the systematic search or from other sources that met the requirements were included in the study.

### 2.2. Sources of Information

The manuscript search was carried out in four databases (*Web of Science, Scopus, SportDiscus and Pubmed*) in January 2020. In general, the search terms were divided into four groups of words as follows: (1) Small-sided games OR Game-based training OR Game-based approaches OR modified games OR task constraints OR conditioned games, (2) Sports initiation OR Young OR Youth OR Children OR Junior OR teenager, (3) Work methodology OR pedagogy OR tactical behaviours OR Tactical OR Tactical skills OR Technical-tactical OR Team behaviour OR Tactical performance OR Tactics OR Procedural Knowledge OR Tactical assessment OR Tactical patterns OR Teaching games for understanding OR Game Sense OR Play Practice OR Games Concept Approach OR Tactical decision learning model OR Sport education OR Tactical games approach OR perception OR Decision making OR execution, and (4) NOT (Physical-fitness OR Fitness level OR Physical condition OR Strength OR Agility OR Endurance OR Balance OR Flexibility OR Speed OR Coordination). Once the search was performed, the results were exported to EndNote Web and duplicates were removed.

### 2.3. Study Selection and Data Extraction Process

After the manuscript search, the title and abstract of each result were screened in order to find potentially relevant studies and to exclude those that did not meet the inclusion criteria.

Four previously designed templates were used for data extraction from the selected papers. The most common elements in studies involving SSGs were considered: (1) number of players, (2) size of pitch area, (3) rule manipulation, and (4) other variables (e.g., participants’ age). To reduce selection bias, each manuscript was independently reviewed by two of the authors (C.F.-E. and M.T.A.R.), who mutually determined whether or not they met basic inclusion criteria. If a consensus could not be reached on inclusion of a study, the matter was settled by consultation with a third author (F.J.G.F.-G.).

### 2.4. Quality Assessment

Quality assessment of selected papers was done using the tool of standard assessment Qualsyst from Kmet et al. [27], which was used previously in other systematic reviews [28,29]. For quantitative studies, this tool includes 14 items which are linked with aspects such as design of the investigation, the sample, the methodology, the data analysis, the results and the conclusion. Each criterion could be punctuated with 2 (satisfactory), 1 (partially satisfactory), (not satisfactory), and NA (not applicable). The final score is obtained through the following formula [(“satisfactory numbers” × 2) + (“partially numbers” × 1)/28-not applicable numbers × 2]. The findings are expressed as percentage from 0% to 100%. The cut-point selected for article inclusion was conservative [27]. Two of the authors (C.F.-E. and M.T.A.R) assessed the quality of each study independently. Discrepancies were solved by a third author (F.J.G.F.-G.)

## 3. Results

### 3.1. Study Selection

The initial search yielded 451 results. The documents were analysed and 20 additional studies were identified in their references. Subsequently, duplicates were removed, with 197 studies excluded. From the remaining 274, 185 were found in full-text version. Following a deeper analysis, 47 studies met the eligibility criteria and were included in the results of this review (see Figure 1).

### 3.2. Quality of Studies

Quality scores of each study were expressed as percentage of maximum quality score in the Table 1, Table 2, Table 3 and Table 4. The percentages ranged from 0.70 to 0.95 (see Appendix A).

### 3.3. Characteristics of the Studies

The main characteristics of the selected studies are presented below (see Table 1, Table 2, Table 3 and Table 4).

## 4. Discussion

The use of small-sided situations allows coaches of all levels to teach technical skills and tactical behaviours [9,10]. The aim of this study was to conduct a systematic review in order to analyse and describe the methodological possibilities that SSGs can provide to youth sport coaches. In this sense, the change in number of players during SSGs is an influencing factor on technique (Table 1, Table 2 and Table 3). In general, the majority of studies agree that reducing the number of players leads to an increase in technical actions. In this respect, it is important to note that this was the frequency of technical elements per player. In this sense, Clemente and Rocha [66] analysed the effect of the number of players in handball and concluded that reducing it (e.g., from 4 vs. 4 to 2 vs. 2) increased the number of ball contacts, interceptions and dribbles per player. In another team sport like basketball, Klusemann et al. [61] confirmed that changing from 4 vs. 4 to 2 vs. 2 increased the probability of performing individual technical actions by up to 60%. This idea was also supported by the studies conducted by Clemente et al. [59] and Conte et al. [62], in which a reduction in the number of players enhanced the playing volume, on one hand, and specific technical actions like the pass, dribble and shot, on the other. Timmerman et al. [53,65] proved in a study on SSGs in hockey that reducing the number of players creates a positive environment that fosters decision making and, therefore, more passes and skilled actions are performed correctly. These data agree with the findings of Sgrò et al. [12] in football.

The majority of manuscripts selected for this study have conducted research in football. In this sport, the results are in line with the aforementioned ones. The works by Jones and Drust [47], Owen et al. [51], Díaz-Cidoncha et al. [38] and Clemente et al. [56] verified that the lower the number of players, the higher the number of technical actions (passes, dribbles and shots) and contacts with the ball. With regard to technical actions, Katy and Kellis [45] and Owen et al. [21] proved that the higher the number of players, the more long passes and headers were performed, on one hand, and the fewer short passes, shots, dribbles and tackles, on the other. In fact, these results are very similar to those found by Da Silva et al. [44] concerning dribbles, shots and goals. In this regard, Sánchez-Sánchez et al. [35] agreed that the lower the number of players, the higher the number of dribbles. Castelao et al. [39] also reported that reducing the number of players had a positive effect on other actions, like defensive balance and penetration. All these results correspond to SSGs with equal number of players. It is worth mentioning the study by Evangelos et al. [43], in which the influence of unevenness in the number of players in 3 vs. 3 and 4 vs. 4 game formats was analysed. For this purpose, a wildcard player with two possible roles (offensive and defensive) or a permanent supernumerary attacking situation with an additional player (e.g., 4 vs. 3) were used. The results revealed that a greater number of interceptions, dribbles and ball receptions were performed when the number of players was increased permanently, maybe due to increased interactions between attackers and defenders. By contrast, in the game format where a wildcard player played always an attacking role when a team recovered the ball, a higher number of passes and turns were completed, perhaps because “numerically superior settings represent an important for the construction of game models based on an organized attack and on the development of principles related to an offensive construction” [76] (p. 10). Nevertheless, the effect of the number of players on tactics was only analysed in the study by Castelao et al. [39] and no significant differences were found. 

Table 1, Table 2 and Table 3 contain the studies that have examined the size of the pitch area as a determining factor of technical–tactical aspects in SSGs. No significant differences were found in any of the variables of the studies by Owen et al. [51] in football and Gabbett et al. [67] in rugby. Nevertheless, there are other studies that did report effects of modifying the size of the playing area. For example, in football, the number of actions like the pass, dribble, and getting away from the defender, increased as the pitch size was reduced [30,42]. In basketball, it was verified that a higher number of technical actions were performed on a smaller pitch area [61]. In this respect, Sgrò et al. [12] found similar results in their study in football. In hockey, it was found that increasing the size of the pitch area led to a higher number of successful dribbles [64]. Besides, several studies have proved that the larger the pitch area, the greater the space among players and the larger the effective playing area [30,34]. The same was confirmed by Castellano et al. [32], in a study on the influence of the pitch length on collective tactical behaviour. The results revealed that intra-team (team width) and inter-team (length of both teams together) variables increased when the size of the pitch area was increased. Therefore, manipulations in field dimensions can influence the area covered by the teams [36]. This may be due to the fact that the size of the pitch influences the players’ perception of space, conditioning the occupation and use of it, as well as the distances between players and their interactions.

Furthermore, the modification of certain rules is another determining factor on technical–tactical aspects and tactical behaviour (Table 1 and Table 2). In this way, the manipulation of constraints, like the rules of the game, can be an effective resource to facilitate learning [77]. In general, when the aim of the game (e.g., to keep possession) was modified, more positional attacks and contacts with the ball occurred [33,49]. By contrast, when the standard rules were applied, there were faster attacks, fewer ball contacts and more individual actions [33]. On the other hand, the use of different type of goals influences the tactical behaviour of players [78]. In this regard, Serra-Olivares et al. [68] reported that in a goal-scoring SSG format, the success rates of decision making and execution were higher than in a possession game format. Other studies have analysed the effect of changing the way of scoring, like, for example, conducting the ball across an imaginary line between two cones or altering the number of goals. Almeida et al. [31] concluded that the line goal format increased the odds of regaining possession through a tackle and decreased the odds of successful interceptions. Moreover, changing from two to eight goals (four per team) made it easier to keep possession and to advance with the ball [36]. Another study on rule manipulation was the one by Almeida et al. [48], in which the number of touches per player (up to two) and the requirements to score (a minimum of four passes) were modified. The results showed that a lower number of touches led to a higher number of shots and that establishing a minimum of mandatory passes enhanced ball possession.

Rule manipulation is typically used by coaches [9] and specific adjustments are made to the design of SSGs in order to increase the tactical or technical load [13]. In this regard, almost all the previous studies regarding rule modifications involved football. For example, the use of the double-tap rule in football had a positive impact on player behaviour, indicating that it can help to improve tactical skills [55]. Only the study by Conte et al. [62], which focuses on dribbling in basketball, has been found to involve other sport modalities. As was described, the number of players, the size of the pitch area and rule manipulation are variables that influence the technical–tactical aspects of the teaching process [2]. Nevertheless, there are more determining factors (Table 4). In this regard, several manuscripts have been found in which the effect of game duration in SSGs is analysed. Klusemann et al. [61] researched the differences between two 5-minute games with a 30-second break of active recovery between them and four 2.5-minute games interspersed with 1-minute breaks of active recovery. However, no significant differences were found. Christopher et al. [70] and Conte et al. [60] examined the differences between playing in a continuous format or with interruptions. In football, a higher number of passes was observed in the continuous playing regime, while, in basketball, a higher number of dribbles were recorded under this format. Summarizing the above, “a significant effect on the playing strategies used by the teams seem to be obtained by manipulating the rules of the SSG exercises and this factor seems to be adequate for supporting the teaching process of a new playing behaviour” [12] (p. 16).

Other variables found in the documents analysed are the participants’ age and years of experience. In general, older participants showed a higher level of collective tactical behaviour [50,72,74], while less experienced participants presented a lower level of tactical behaviour [69] and performed shorter offensive sequences with more individual actions [48]. These results regarding age could have a relationship with the studies by González-Víllora et al. [75] and González-Víllora, García-López [73], who proved that players acquired procedural knowledge earlier than declarative or theoretical knowledge. Da Silva et al. [44] examined whether the participants’ maturation level had an effect on technical aspects, but no significant correlations were found.

Lastly, there was also one study in which the effects of the coach’s presence or absence were analysed [71]. It was concluded that the coach’s presence made players stressed or uncomfortable, having a negative effect on technical aspects like successful passing.

It must be acknowledged that this study presents some limitations. Despite the large amount of results obtained, most studies involve football, making it difficult to extrapolate some of the results to other team sports. Furthermore, some of the manuscripts included in the present review were taken from other sources, suggesting that there are more studies that could be included.

## 5. Conclusions

The results seem to indicate that SSGs can be used as an interesting methodological resource to work on technique and tactics in team sports at young ages. To do so, it is necessary to establish the aim to be achieved and which variables should be modified and how, some of the most determining ones being the number of players, the size of the pitch area and the manipulation of certain rules.

Once the most relevant aspects of SSGs related to the teaching of technical–tactical elements with young athletes in team sports have been analysed, new research lines appear that can shed more light on this topic. For example, it would be interesting to prepare research proposals based on scientific evidence that include proper training programmes and progression with SSGs according to age group. Moreover, it seems necessary to analyze studies in which an intervention is carried out using SSGs, measuring before and after the intervention, as well as comparing the control and experimental groups.

To conclude, a summary of the results of this systematic review is presented below, and some possible practical implications are shown when teaching sports at young ages are proposed (Table 5). The summary sometimes makes reference to one study with one age group, so its implications should be taken with caution, since the inclusion of younger or older players alters the results.

## Figures and Tables

**Figure 1 ijerph-17-01884-f001:**
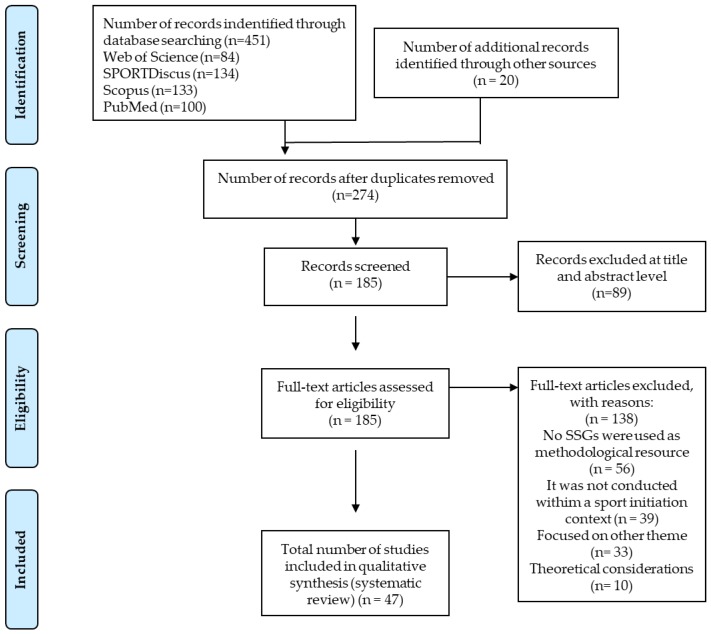
Flow chart.

**Table 1 ijerph-17-01884-t001:** Effects of small-sided games (SSGs) on technical-tactical aspects in football.

Author/s [Sport]	*N* [gender]	Age	Type of SSG	Size (m)	t [B] (min)	Quality Score %	Results
Mallo and Navarro, [14]	10 [M]	18.6	3 vs. 3 3 vs. 3 +2EW G+3 vs. 3+G	33 × 20	3 × 5′ [20′]	85	Players completed more ball contacts in the possession format than in the other two. The number of short passes was higher than in the third game format and the percentage of wrong passes was higher than in the second game format.
Olthof et al. [30]	148 [NA]	12.5 to 17.9	G+4 vs. 4+G	40 × 30 68 × 47	4′ [4′]	95	There were a higher number of transitions, set pieces and shots on a small pitch. On a large pitch, intra- and inter-team distances were longer.
Almeida et al. [31]	16 [M]	12.61 14.86	4 vs. 4	30 × 20	10′ [5′]	90	Line goal mode increased the odds of regaining possession through tackle and decreased the odds of successful interceptions. Double goal mode decreased the odds of regaining possession through turnover and long plays.
Castellano et al. [32]	14 [M]	13.5 and 14.3	G+6 vs. 6+G	30 × 40 40 × 40 50 × 40 60 × 40	4 x 7′ [4′]	85	Some intra-team (e.g., team length) and inter-team (e.g., distance between centroids) variables increased when the pitch area increased.
Machado et al. [33]	14 [NA]	13.82	G+6 vs. 6+G	52 × 32	30′	80	The small-sided keeping possession game induced positional attacks, a higher number of players involved and greater use of the side areas. The goal scoring game induced faster offensive sequences, long passes and individual behaviours.
Silva, Duarte et al. [34]	20 [M]	16.3	G+4 vs. 4+G	36.8 × 23.8 47.3 × 30.6 57.8 × 37.4	7′ [7′]	90	The larger the pitch, the larger the effective playing area and the larger the area the team occupies.
Sánchez-Sánchez et al. [35]	22 [M]	17.2	4 vs. 4 G+4 vs. 4+G (2IW or 2EW)	30 × 40	4′[2′]	85	Greater number of dribbling situations during 4 vs. 4 with no goalkeeper compared with 4 vs. 4 with no goalkeeper but with internal or external wildcards. Greater number of successful actions compared with 4 vs. 4 with goalkeeper.
Serra-Olivares, González-Víllora, and García-López [36]	21 [NA]	8 to 9	3 vs. 3	20 × 30	2 x 4′ [2′]	75	In the game format with higher number of goals, greater decision making (no significant difference) and better tactical adaptation were observed, since it was more difficult to keep possession and to advance to the goal in the standard game.
Serra-Olivares, González-Víllora, García-López, and Araújo [37]	21 [NA]	8.7	3 vs. 3	22 × 32 29.5 × 15	2 × 4′	75	No significant differences were found in the technical variables under study between the two types of game.
Díaz-Cidoncha et al. [38]	54 [M]	NA (U9 and (U14)	5 vs. 5 7 vs. 7 9 vs. 9	20 × 30 30 × 45 45 × 60	20′	80	The lower the number of players, the higher the number of touches per player and goalkeeper. The number of attacking plays, dribble and pass attempts was also higher in the 5 vs. 5 format.
Castelao et al. [39]	10 [NA]	NA (U11)	G+3 vs. 3+G G+5 vs. 5+G	36 × 27 60 × 45	8′	75	In the 3vs3 format, higher scores were obtained in penetration, defensive coverage, shots at goal and shots at own goal, while higher scores were achieved in offensive unity and balance in the 5 vs. 5 format. No tactical differences were found.
Silva, Garganta et al. [40]	18 [NA]	NA (U11)	G+3 vs. 3+G G+6 vs. 6+G	30 × 19.5 60 × 39	8′	75	Players displayed safer behaviours in larger formats and more aggressive ones in smaller formats.
Abrantes et al. [41]	16 [M]	15.75	3 vs. 3 4 vs. 4	20 × 30 20 × 40	4 × 4′ [6′]	85	The results were identical in both formats.
Casamichana and Castellano [42]	10 [M]	15.5	G+5 vs. 5+G	62 × 44 50 × 35 32 × 23	8′ [5′]	90	Most actions under study increased as the size of the pitch area was reduced.
Evangelos et al. [43]	9 [M]	17.2	3 vs. 3 3 vs. 3+W 4 vs. 3 4 vs. 4 4 vs. 4+W 5 vs. 4	20 × 25 25 × 30	4 × 3′ [12′]	85	A higher number of interceptions, dribbles and receives occur in supernumerary situations. More passes and turns are completed with an equal number of players or an additional offensive player.
Da Silva et al. [44]	16 [M]	13.5	3 vs. 3 4 vs. 4 5 vs. 5	30 × 30	3 × 4′ [9′]	90	There were a greater number of dribbles, crosses, shots and goals in the smallest format.
Katis and Kellis [45]	34 [NA]	13	G+3 vs. 3+G G+6 vs. 6+G	15 × 25 30 × 40	10 × 4′ [9′]	92	The number of short passes, kicks, dribbles, tackles and goals was higher in the smaller format. More long passes and headers were performed in the larger format.
Kelly and Drust [46]	8 [NA]	18	G+5 vs. 5+G	30 × 20 40 × 30 50 × 40	16′ [8′]	92	The smaller the size, the higher the number of tackles and shots.
Jones and Drust [47]	8 [M]	7	4 vs. 4 8 vs. 8	30 × 25 60 × 40		90	A reduction in the number of players increased the number of ball contacts per player.
Almeida et al. [48]	8 [M]	12.8	G+3 vs. 3+G	40 × 30	3 × 10′ [15′]	85	In the 2-touch game format, a higher number of goals and shots on goal and a faster playing pattern were recorded. The 4-passes-to-score rule led to greater ball possession.
Rebelo et al. [49]	10 [M]	17.2	5 vs. 5 G+5 vs. 5+G	30 × 20 40 × 30 50 × 40	2 × 10′ [10′]	85	The ball-possession game induced a higher number of passes and ball touches on all pitch sizes. Fewer errors were made during the goal-scoring game on the small and medium pitch sizes.
Serra-Olivares et al. [50]	21 [NA]	8 to 9	3 vs. 3	30 × 22 20 × 20	2 × 4′ [3′]	70	Decision-making and execution were more successful in the goal-scoring game.
Owen et al. [51]	NA	17.4	1 vs. 1 2 vs. 2 3 vs. 3 4 vs. 4 5 vs. 5	5 × 10; 10 × 15; 15 × 20 10 × 15; 15 × 20; 20 × 25 15 × 20; 20 × 25; 25 × 30 20 × 25; 25 × 30; 30 × 35 25 × 30; 30 × 35; 35 × 40	9′ [24′]	75	An increase in the number of players led to a decrease in the number of technical actions per player. No significant difference was found in any of the actions under study.
Machado et al. [52]	268 [M]	16.49	G+3 vs. 3+G	27 × 26	1 × 4′	90	In 4 vs 4 format, players realise a similar quantity of tactical actions regardless of the positional role. However, the quality of tactical behaviour was significantly affected by the positional role.
Moreira et al. [53]	36 [NA]	13.7	3 vs. 3 3 vs. 3 +1 3 vs. 3 + 1	36 × 27 36 × 27 40 × 29	1 × 4′ [4′]	85	The reduction in the relative and absolute playing area elevated the frequency of offensive unity and the level of interaction between players.
Práxedes et al. [54]	19 [NA]	10.63	5 vs. 5 + 1W 3 vs. 3 + 2W 4 vs. 3 G+4 vs. 4 +G +1W	15 × 10 30 × 20	[NA]	85	The nonlinear pedagogy intervention programme improved the decision making and the execution.
Sousa et al. [55]	36 [NA]	15.13	3 vs. 3	36 × 27	1 × 4′[4′]	90	Two-touch rule increased the ball circulation and reduced the tactical complexity of the defensive performance.
Clemente et al. [56]	16 [NA]	10.1	3 vs. 3 6 vs. 6	15 × 20 30 × 22	3 × 3 [2′] 3 × 6′ [2′]	80	The smaller format increased significant the number of individual technical actions.
Folgado et al. [57]	20 [NA]	14.1	G+4 vs. 4+G	30 × 40 40 × 30	1 × 6′[3′]	85	In the 30 × 40m pitch, results showed a lower distance between team centroids, higher number of shots and lateral passes. In the 40 × 30m pitch the players which covered more distance at higher intensities presented more passes and dribbles.
Machado et al. [58]	20 [M]	13.5 16.3	G+3 vs. 3+G G+4 vs. 4+G	36 × 27 47.72 × 29.54	3 × 10′[10′]	85	In maintaining ball possession games, younger players presented greater difficulties in smaller format. The bigger format can be used in younger players to improve the tactical performance in progression to target games and representative games.

M: Male; F: Female; G: Goalkeeper; B: Break; EW: External wildcard player; IW: External wildcard player; NA: Not available; m: metres; min: minutes.

**Table 2 ijerph-17-01884-t002:** Effects of SSGs on technical-tactical aspects in basketball.

Author/s [Sport]	*N* [gender]	Age	Type of SSG	Size (m)	t [B] (min)	Quality Score %	Results
Clemente et al. [59]	10 [M]	14.75	2 vs. 2 + 2W 3 vs. 3 + 2W 4 vs. 4 + 2W	15 × 11 19 × 13 22 × 15	5′ [3′]	85	Smaller formats led to greater playing volume, number of attacks with ball and efficiency index and better score.
Conte et al. [60]	21 [M]	15.4	2 vs. 2 4 vs. 4	28 × 15	3 × 4′ [2′] 3 × 7′ [1′]	90	The 2vs2 format showed higher number of dribbles, passes, shots and turnovers compared with 4 vs. 4.
Klusemann et al. [61]	8[M] 9[F]	17.4 and 18.2	2 vs. 2 4 vs. 4	28 × 15 14 × 7.5	4 × 2.5′ [1′] 2× 5′ [30″]	85	Participants performed ~60% more technical elements (per player) in 2 vs. 2 than in 4 vs. 4 situations. On a small pitch, ~20% more technical elements (per player) were performed than on a large pitch
Conte et al. [62]	23 [M]	15.5	4 vs. 4	28 × 15	3 × 4′ [2′]	85	The total number of passes, the number of correct and wrong passes and the number of interceptions were significantly higher in the no-dribble game.
Bredt et al. [63]	12 [M]	17.1	3 vs. 3	15 × 14	2 × 5′ [3′]	85	The space creation with ball, dribbled, space creation without the ball, set offenses, and fast breaks have high reliability in the 3 vs. 3 with man-to-man defense in half playing area than with man-to-man defenses in full playing area.

M: Male; F: Female; G: Goalkeeper; B: Break; W: Wildcard player; m: metres; min: minutes.

**Table 3 ijerph-17-01884-t003:** Effects of SSGs on technical-tactical aspects in other sports.

Author/s [Sport]	*N* [gender]	Age	Type of SSG	Size (m)	t [B] (min)	Quality Score %	Results
Timmerman et al. [64] [Hockey]	25 [NA]	12.2	8 vs. 8 11 vs. 11	77 × 47 55 × 46 91 × 55 64 × 54	25′	85	Manipulation of the number of players led to an increase in successful passes, skilled and successful actions and also created an environment that enhanced decision making. An increase in pitch size led to a decrease in unsuccessful dribbles.
Timmerman, Savelsbergh, & Farrow [65] [Hockey]	13	13.2	3 vs. 3 6 vs. 6	28 × 17 24 × 40	2 × 7.5′ [2.5′]	85	Lowering the number of players elevated the technical actions. The possession game increased the number of passes and decreased dribbles and tackles. The two-goals game increased the goals the cage hockey game increased passing.
Clemente & Rocha [66] [Handball]	8 [M]	18.25	2 vs. 2 3 vs. 3 4 vs. 4	10 × 7.5 20 × 7.5	5′	85	The number of touches, dribbles and interceptions per player was higher when a smaller number of players were involved.
Gabbett et al. [67] [Rugby]	32 [NA]	23.6 and 17.3	8 vs. 8	10 × 40 40 × 70	8′	80	No significant difference was found in any of the variables.

M: Male; F: Female; G: Goalkeeper; B: Break; NA: Not available; m: metres; min: minutes.

**Table 4 ijerph-17-01884-t004:** Effects of other variables of SSGs on technical–tactical aspects.

Author/s [Sport]	Variable	*N* [gender]	Age	Type of SSG	Size (m)	t [B] (min)	Quality Score %	Results
Da Silva et al. [44] [Football]	Maturation	16 [M]	13.5	3 vs. 3 4 vs. 4 5 vs. 5	30 × 30	3 × 4′ [9′]	90	No significant differences in technique were found with different maturation levels.
Almeida et al. [48] [Football]	Training experience	28 [M]	12.84 and 12.91	G+3 vs. 3+G G+6 vs. 6+G	46 × 31 62 × 40.04	2 × 10′ [5′]	75	The more experienced players performed longer offensive sequences with greater ball circulation. By contrast, the less experienced players completed faster and more individual offensive sequences.
Klusemann et al. [61] [Basketball]	Duration	8[M] 9[F]	17.4 18.2	2 vs. 2 4 vs. 4	28 × 15 14 × 7.5	4 × 2.5′ [1′] 2 × 5′ [30″]	85	No significant differences in technique were found when modifying the work-to-rest ratio.
Conte et al. [62] [Basketball]	Duration	21 [M]	15.4	2 vs. 2 4 vs. 4	28 × 15	3 × 4′ [2′] 4 × 1′ [1′]	90	The continuous regime revealed higher number of dribbles than the intermittent regime.
Serra-Olivares et al. [68] [Football]	Age and skill level	21 [NA]	8.3	3 vs. 3	22 × 32	2 × 4′ [1′]	80	Performance of older and more skilled players was significantly better in getting-free decisions and in passing decisions to keep the ball possession.
Barnabé et al. [69] [Football]	Years of experience	36 [M]	15.2 16.3 17.4	G+5 vs. 5+G	33 × 60	8′	90	In offensive, defensive and mixed phases, older and more experienced players occupied a greater surface area and showed higher stretch index.
Christopher et al. [70] [Football]	Duration	12 [NA]	15.8	G+5 vs. 5+G	50 × 32	8′ [0′] 2 × 4′ [1′] 4 × 2′ [45″]	85	There were more shots and goals in the 4- and 2-min formats. There were more successful passes in the continuous 8-min format.
Falces-Prieto et al. [71] [Football]	Coach’s presence	27 [M]	17.0	G+6 vs. 6+G	NA	6′ [5′]	90	The percentage of successful passes decreases while the percentage of unsuccessful passes increases in the coach’s presence. The number of successful control-conduction passes increases in the coach’s presence.
Olthof et al. [72] [Football]	Age	39 [M]	15.4 17.4	G+5 vs. 5+G	40 × 30	6′	90	Older players showed significantly higher lateral stretch index and significantly lower length-per-width ratio than younger players.
González-Víllora et al. [73] [Football]	Procedural and declarative knowledge	16 [NA]	14	7 vs. 7	64 × 44	2 × 4′ [3′]		Players acquired procedural knowledge earlier than declarative knowledge. Besides, they performed better at decision making than at execution.
Folgado et al. [74] [Football]	Age	30 [NA]	10.53	G+3 vs. 3+G G+4 vs. 4+G	30 × 20	8′ [6′]	85	Older players showed higher level of collective tactical behaviour.
González-Víllora et al. [75] [Football]	Type of knowledge	14 [NA]	11 to 12	5 vs. 5	52 × 40	2 × 4′ [3′]	75	Players showed greater procedural than declarative knowledge. They performed better at decision making than at execution.

M: Male; F: Female; G: Goalkeeper; B: Break; NA: Not available; m: metres; min: minutes.

**Table 5 ijerph-17-01884-t005:** Summary and possible practical implications of the studies analysed with regard to technical–tactical fundamentals.

Variables	Modifications	Effect	Possible Implications
Number of players	Lower number of players	Increased ball contact per player	Take it into account when working on individual technical aspects (e.g., pass, dribble).
	Higher number of players	Increased number of long passes	The number of players can be increased in order to work on this aspect, both during the offensive and defensive phases.
Size of pitch area	Greater playing area	Larger area occupied by the team and greater distances among players	Increasing the pitch area fosters collective tactical behaviour and game knowledge.
	Smaller playing area	Higher number of technical actions and reduced possession	Coaches can make a training progression to work on individual playing.
Playing rules	Limited number of ball contacts	Higher number of shots due to a faster playing pattern	Design a progression of increasing difficulty to work on play finishing.
	Minimum number of passes required before shooting	Greater possession, players’ involvement and number of passes	Coaches can manipulate this rule to work on more positional than direct attacking sequences.
	Limited dribbling	Higher number of passes and interceptions	This can be used to improve the pass, as well as the defending action to prevent it.
	Keeping possession	More positional attacks, greater use of the area next to the touch lines and higher number of ball contacts per player	Take this rule into account to work on players’ spatial organisation and to increase participation.
	Different ways of scoring	Enhanced ball recovery, possession and advancement with ball	Depending on the way of scoring, different training fundamentals can be worked on.
Duration	Continuous/Intermittent	Continuous playing seems to affect some technical–tactical elements	More studies are needed to provide more data in this regard.
Coach	Presence	Uncomfortable atmosphere that may affect decision making	It is important to know that the coach’s presence may affect the player.

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
