# Peer review of "Small-Sided Games as a Methodological Resource for Team Sports Teaching: A Systematic Review"

_ijerph, 2020, doi:10.3390/ijerph17061884_

Round 1
Reviewer 1 Report
General comments
The article “Small-sided games as a methodological resource for team sports teaching: A systematic review” aims to overview the available literature regarding the manipulations in SSG for tactical and technical training in team sports. The subject is interesting, but when considering soccer, there is a large number of high-quality reviews available in the literature (higher than the current one). When considering other team sports, a systematic review seems like a gap, however, the authors should focus only on those less investigated modalities. Finally, there are serious flaws regarding the article selection, so the current study has no external validity. It can be suitable for publication after major issues, mainly considering focusing on non-reviewed team sports and improving the article selection process.
Specific comments
Abstract
Line 14: replace work by stimuli
Introduction
Line 42: please remove the space.
Lines 47-50: the research problem, according to the authors, is the absence of previous systematic reviews regarding team sports, since all the available literature is focused on soccer (Ometto et al., 2018; Sarmento et al., 2018, for example). However, if we have a lot of available information regarding soccer, why did you include this modality in the current study? Considering the gap you have mentioned, it would be better to include only other team sports.
Line 51: you conducted a literature review or a systematic review? These are two different types of reviews and you have mentioned both (literature review in the abstract and systematic review in the introduction).
Methods
There are serious flaws that make the current research irreproducible. For example, when the articles’ search was done? Depending on the date, there are many missing articles that comprise the eligibility criteria (just as an example, since many articles are missing: Bredt et al., 2018; Lemes et al., 2019; Moreira et al., 2019; Sousa, Bredt, Greco, Manuel Clemente, & Praça, 2019). If you consider the papers published by Israel Teoldo, Gibson Praça and Filipe Clemente, (please, check the Research Gate for more details) you have more than 50 non-selected studies that accomplish with the eligibility criteria.
Since “sport initiation context” is an important characteristic you are looking for in the studies, you must clearly define it in the methods section. Is there a reference age? A competitive level? A time spent in deliberate practice? Some studies selected, in my opinion, are not focused on initiation sport. Just as an example
Clemente, F.P.; Lourenco, F.M.; Mendes, R.; Octavio, P. The effects of Small-Sided Games and Conditioned 180 Games on Hearth Rate Responses, Technical and Tactical performances measured by mathematical 181 Res. J. Applied Sci. 2016, 11, 7-13. doi: 10.3923/rjasci.2016.7.13 - There are U-16 groups playing at the national level;Clemente, F.; Rocha, R. The effects of task constraints on the heart rate responses of students during small-197 sided handball games. Slov. 2012, 18, 27-35. There are U-18 groups.
Since the quality score of the articles was assessed, this should be used as a selection criterion (poor studies methodologically should be removed).
Results
There is a serious flaw regarding the interpretation of the data of technical actions. To compare studies with different bout durations and number of players in regard, for example, to the number of passes, you must relativize the data for the time (a minute, for example) and for each athlete. For example, a game may present a higher number of actions per player per minute (a reduced one) than a larger format, but present the same number of total actions than the larger. Mathematically talking, if the number of passes per minute per player in a 3v3 game is 3, and this number is 1 in a 9vs9 game if both games have the same time, the total number of passes registered in both games is equal (=18). So, the conclusion “the majority of studies agree that reducing the number of players leads to an increase of technical actions” is only true when talking about the relative number, no the absolute one (and you did not relativize the numbers in the current review study).
Discussion
There is a misunderstanding regarding this section. Here, you mostly summarized the results, not discussing them. Looking for justifications is the main aim of this section, which has not been addressed by the authors. For example (just as an example, since this can be seen in the whole section), lines 32-33 present just a result (with no rationale).
Another example of the abovementioned issue is “The results revealed that intra-team (team width) and inter-team (length of both teams together) variables increased when the size of the pitch area was increased”. What are the reasons for this result? How do you explain it? In its current form, this is not a discussion, it is just a summary of the results.
Line 61: this is not true. There is at least one study (Bredt et al., 2018) in basketball in which rules were modified. Again, the articles’ selection criteria are not clear.
Conclusions
Lines 88-89: this conclusion is not supported by the results. To conclude this, you should have selected longitudinal studies instead of transversal ones, in which the tactical skills are assessed previously and after a period of time with small-sided games (for example, like was done in this study: Práxedes, Del Villar Álvarez, Moreno, Gil-Arias, & Davids, 2019).
You have not enough evidence for many variables presented in table 5. Many of them were investigated in a few studies (some in just one) and in only one modality (often soccer). So, this is a speculative table, no a conclusion.
REFERENCES
Bredt, S. G. T., Morales, J. C. P., Andrade, A. G. P., Torres, J. O., Peixoto, G. H., Greco, P. J., … Chagas, M. H. (2018). Space Creation Dynamics in Basketball Small-Sided Games. Perceptual and Motor Skills, 125(1). https://doi.org/10.1177/0031512517725445
Lemes, J. C., Luchesi, M., Diniz, L. B. F., Bredt, S. D. G. T., Chagas, M. H., & Praça, G. M. (2019). Influence of pitch size and age category on the physical and physiological responses of young football players during small-sided games using GPS devices. Research in Sports Medicine, 1–11. https://doi.org/10.1080/15438627.2019.1643349
Machado, G., Bach Padilha, M., González Víllora, S., Clemente, F. M., & Teoldo, I. (2019). The effects of positional role on tactical behaviour in a four-a-side small-sided and conditioned soccer game. Kinesiology, 51(2), 261–270. https://doi.org/10.26582/k.51.2.15
Moreira, P. E. D., Barbosa, G. F., Murta, C. D. C. F., Morales, J. C. P., Bredt, S. D. G. T., Praça, G. M., & Greco, P. J. (2019). Network analysis and tactical behaviour in soccer small-sided and conditioned games: influence of absolute and relative playing areas on different age categories. International Journal of Performance Analysis in Sport, 20(1), 1–14. https://doi.org/10.1080/24748668.2019.1705642
Ometto, L., Vasconcellos, F. V., Cunha, F. A., Teoldo, I., Souza, C. R. B., Dutra, M. B., … Davids, K. (2018). How manipulating task constraints in small-sided and conditioned games shapes emergence of individual and collective tactical behaviours in football: A systematic review. International Journal of Sports Science & Coaching, 13(6), 1200–1214. https://doi.org/10.1177/1747954118769183
Práxedes, A., Del Villar Álvarez, F., Moreno, A., Gil-Arias, A., & Davids, K. (2019). Effects of a nonlinear pedagogy intervention programme on the emergent tactical behaviours of youth footballers. Physical Education and Sport Pedagogy, 1–12. https://doi.org/10.1080/17408989.2019.1580689
Sarmento, H., Clemente, F. M., Harper, L. D., Costa, I. T. da, Owen, A., Figueiredo, A. J., … Figueiredo, A. J. (2018). Small sided games in soccer – a systematic review. International Journal of Performance Analysis in Sport, 18(5). https://doi.org/10.1080/24748668.2018.1517288
Sousa, R. B. e, Bredt, S. T., Greco, P. J., Manuel Clemente, F., & Praça, G. M. (2019). Influence of limiting the number of ball touches on players’ tactical behaviour and network properties during football small-sided games. International Journal of Performance Analysis in Sport, 19(6), In press.
Author Response
COVER LETTER
Manuscript ID: ijerph-714510. Type of manuscript: Article. Title: Small-sided games as a methodological resource for team sports teaching: A systematic review
|
Reviewer 1’s comments and suggestions for authors |
Details of the revisions and responses |
|
Line 14: replace work by stimuli. |
We have replaced work by stimuli . |
|
The research problem, according to the authors, is the absence of previous systematic reviews regarding team sports, since all the available literature is focused on soccer (Ometto et al., 2018; Sarmento et al., 2018, for example). However, if we have a lot of available information regarding soccer, why did you include this modality in the current study? Considering the gap you have mentioned, it would be better to include only other team sports. |
Dear reviewer, we thank you for your suggestion, which we consider interesting and worthy of being taken into account for future research. However, we have considered including soccer studies because our intention is to give a more complete vision of the aspects that can be worked through the reduced games and to analyze its incidence in all the sports of a same category. Otherwise, since there is little research on other sports, vision would be reduced and less transferable to similar sports. |
|
You conducted a literature review or a systematic review? These are two different types of reviews and you have mentioned both (literature review in the abstract and systematic review in the introduction). |
We have conducted a systematic review. Now, we mention “systematic review” throughout the manuscript. |
|
Depending on the date, there are many missing articles that comprise the eligibility criteria (just as an example, since many articles are missing: Bredt et al., 2018; Lemes et al., 2019; Moreira et al., 2019; Sousa, Bredt, Greco, Manuel Clemente, & Praça, 2019). If you consider the papers published by Israel Teoldo, Gibson Praça and Filipe Clemente, (please, check the Research Gate for more details) you have more than 50 non-selected studies that accomplish with the eligibility criteria. |
The manuscript search was carried out in four data bases (Web of Science, Scopus, SportDiscus and Pubmed) in May and June 2018. Perhaps, these studies had not been published or indexed yet.
We have not searched Research Gate, but we will consider it in future investigations. Thank you for your recommendation. |
|
Since “sport initiation context” is an important characteristic you are looking for in the studies, you must clearly define it in the methods section. Is there a reference age? A competitive level? A time spent in deliberate practice? Some studies selected, in my opinion, are not focused on initiation sport. Just as an example Clemente, F.P.; Lourenco, F.M.; Mendes, R.; Octavio, P. The effects of Small-Sided Games and Conditioned 180 Games on Hearth Rate Responses, Technical and Tactical performances measured by mathematical 181 Res. J. Applied Sci. 2016, 11, 7-13. doi: 10.3923/rjasci.2016.7.13 - There are U-16 groups playing at the national level; Clemente, F.; Rocha, R. The effects of task constraints on the heart rate responses of students during small-197 sided handball games. Slov. 2012, 18, 27-35. There are U-18 groups. |
We have defined “sport initiation context” in the methods section. |
|
Since the quality score of the articles was assessed, this should be used as a selection criterion (poor studies methodologically should be removed). |
The cut-point selected for article inclusion was conservative (<.70). This Thresholds is relatively robust (kmet et al. 2004). |
|
Mathematically talking, if the number of passes per minute per player in a 3v3 game is 3, and this number is 1 in a 9vs9 game if both games have the same time, the total number of passes registered in both games is equal (=18). So, the conclusion “the majority of studies agree that reducing the number of players leads to an increase of technical actions” is only true when talking about the relative number, no the absolute one (and you did not relativize the numbers in the current review study). |
We have relativized the numbers about this issue. |
|
There is a misunderstanding regarding this section. Here, you mostly summarized the results, not discussing them. Looking for justifications is the main aim of this section, which has not been addressed by the authors. For example (just as an example, since this can be seen in the whole section), lines 32-33 present just a result (with no rationale). Another example of the abovementioned issue is “The results revealed that intra-team (team width) and inter-team (length of both teams together) variables increased when the size of the pitch area was increased”. What are the reasons for this result? How do you explain it? In its current form, this is not a discussion, it is just a summary of the results. |
We have looked for possible explanations to the results found. |
|
Lines 88-89: this conclusion is not supported by the results. To conclude this, you should have selected longitudinal studies instead of transversal ones, in which the tactical skills are assessed previously and after a period of time with small-sided games (for example, like was done in this study: Práxedes, Del Villar Álvarez, Moreno, Gil-Arias, & Davids, 2019). |
We have expressed the conclusions with caution. |
|
You have not enough evidence for many variables presented in table 5. Many of them were investigated in a few studies (some in just one) and in only one modality (often soccer). So, this is a speculative table, no a conclusion. |
We have expressed the possible practical implications with caution. |
THANK YOU for your comments and suggestions.

Reviewer 2 Report
Principio del formulario
Final del formulario
COMMENTS FOR THE AUTHOR:
The article and the different parts are well organized. The authors arrive at the conclusion that small-sided games are effective methodological resource for team sports teaching. The paper is within the scope of the journal and deals with a relevant topic. However, I have some concerns with regard to this meta-analysis, which need to be addressed before publication might be recommended.
Abstract
You start speaking about methodological SSG possibilities as a physical, tec-tactical & tactical interaction (see page 1 lines 13 & 14), but during the text (discussion) it´s only addressed the tec-tac actions, what about physical & tactical behaviour implications?
Introduction
Please change the start of the second paragraph “Thanks to the contributions…”, the definition of small-side game has not been proposed the first time in these papers. Change the redaction.
Lines 48 and 49, when you have reported changes in ....., please try to differentiate technical and team behaviour variables (that is, shots, passes, dribbles… and surface area, stretch index, width, length…).
The objective appears in the last paragraph and the objective of the first paragraph in Materials and Methods section it is not the same. That is the porpoise of the paper and the porpoise of the systematic revision are not the same. Please review.
Page 2, line 82. What is the meaning of four previously templates? I think this needs more explanation.
Page 2, line 83. There is nothing in the introduction about this: “As explained in the introduction, the most common elements in studies involving SSGs were considered: 1) number of players, 2) size of pitch area, 3) rule manipulation, and 4) other variables (e.g., participants’ age).”
Results
Please include the table with the paper quality assessment.
Tables 1, 2 and 3. Please separate selected studies considering sport and not changes in rules. I think could be more efficient. In this way you are not duplicating information (repeating studies) when you are describing the proposals. Please, detail the connection between players’ number and dimension (maybe it could be interesting to include the relative space per player).
Each table in the results need to be explained a little more, deeply.
In addition, for the description of the studies, I consider pitch orientation & presence/absence and the type of goals (Regular goals, F7 goals, small goals, scoring goal through stop ball procedure) are interesting to be included.
Discussion
An introduction paragraph it is necessary in this section.
Please include some information about de quality of the papers.
Please check the lines 2 and 3 in the discussion section: “The increase of tec-tac action are linked with an increase in the number of successful /unsuccessful actions.” more or less mistakes during de SSG?
The first two paragraph the authors argue very simple topic “less players in the field more touches per player”. This is a conclusion very logical. Please try to change or reduce this simplistic information.
Please check the numbers of the cites, there are some mistakes.
Lines 18 and 19, maybe it´s due to the Effective Area of Play (EAP) more than the number of players? Please check it.
In line 32, “The effect of the number of players on tactics was only analysed in the study by Castelao et al. [32] and no significant differences were found.”. These two papers also study this: 1) Olthof, S. B. H., Frencken, W. G. P., & Lemmink, K. A. P. M. (2017). Match-derived relative pitch area changes the physical and team tactical performance of elite soccer players in small-sided soccer games. Journal of Sports Sciences, 0(0), 1–7. https://doi.org/10.1080/02640414.2017.1403412, 2) Silva, P., Vilar, L., Davids, K., Araújo, D., & Garganta, J. (2016). Sports teams as complex adaptive systems: manipulating player numbers shapes behaviours during football small-sided games. SpringerPlus, 5(1), 1–10. https://doi.org/10.1186/s40064-016-1813-5
In addition, the summary and practical implications that you present in the table 5 sometimes make reference to an only study with an only group of age, so it could be a limitation. Maybe the same study with younger or older players differ the results. Also, you don´t add information about if the sampling included high talented young players or recreational level.
Other simplistic conclusion “The results showed that limiting the use of the dribble increased the number of passes and interceptions”. Please try to rewrite this sentence.
Conclusion
The first paragraph must be changed of section. The adequate space for it is the first paragraph of the discussion section.
The limits of the study must not be in conclusion section. Furthermore, if the manuscript included in the present study review were taken from other sources, it means there was something wrong in the searching strategy. Maybe is necessary to repeat the revision with new term and or synonymous.
Author Response
COVER LETTER
Manuscript ID: ijerph-714510. Type of manuscript: Article. Title: Small-sided games as a methodological resource for team sports teaching: A systematic review
|
Reviewer 2’s comments and suggestions for authors |
Details of the revisions and responses |
|
You start speaking about methodological SSG possibilities as a physical, tec-tactical & tactical interaction (see page 1 lines 13 & 14), but during the text (discussion) it´s only addressed the tec-tac actions, what about physical & tactical behaviour implications? |
We speak in discussion about tactical behaviour (see page 12, line 46; page 12, lines 52, 81, 82). Table 5 shows possible implications on tactical behaviour. |
|
Please change the start of the second paragraph “Thanks to the contributions…”, the definition of small-side game has not been proposed the first time in these papers. Change the redaction. |
We have changed the redaction of the start of the second paragraph. |
|
Lines 48 and 49, when you have reported changes in ....., please try to differentiate technical and team behaviour variables (that is, shots, passes, dribbles… and surface area, stretch index, width, length…). |
We have differentiated technical aspects. |
|
The objective appears in the last paragraph and the objective of the first paragraph in Materials and Methods section it is not the same. That is the porpoise of the paper and the porpoise of the systematic revision are not the same. Please review. |
We have reviewed this issue. |
|
Page 2, line 82. What is the meaning of four previously templates? I think this needs more explanation. |
The data extraction process was realised using four different templates which were based in the most common elements in studies involving SSGs were considered. |
|
Please include the table with the paper quality assessment. |
We have included this table in Appendix A. |
|
Page 2, line 83. There is nothing in the introduction about this: “As explained in the introduction, the most common elements in studies involving SSGs were considered: 1) number of players, 2) size of pitch area, 3) rule manipulation, and 4) other variables (e.g., participants’ age).” |
We have reviewed this issue. |
|
Discussion An introduction paragraph it is necessary in this section. |
We have included an introduction paragraph. |
|
Please include some information about de quality of the papers. |
We have included more information in lines 112 and 113 in the results section. |
|
Tables 1, 2 and 3. Please separate selected studies considering sport and not changes in rules. I think could be more efficient |
We have separated the tables 1, 2 and 3 considering sport. |
|
In addition, the summary and practical implications that you present in the table 5 sometimes make reference to an only study with an only group of age, so it could be a limitation. |
We have included this paragraph: “The summary sometimes make reference to an only study with an only group of age, so the implications should be taken with caution, since with younger or older players differ the results”. |
|
Please check the numbers of the cites, there are some mistakes. |
We have checked the numbers of the cites. |
|
Other simplistic conclusion “The results showed that limiting the use of the dribble increased the number of passes and interceptions”. Please try to rewrite this sentence. |
We have removed this sentence. |
|
Also, you don´t add information about if the sampling included high talented young players or recreational level. |
We have removed this information. |
|
Conclusion The first paragraph must be changed of section. The adequate space for it is the first paragraph of the discussion section. |
We have changed this paragraph. |
|
The limits of the study must not be in conclusion section. |
We have changed the limits of the study from conclusion section to discussion section. |
THANK YOU for your comments and suggestions.

Round 2
Reviewer 1 Report
Dear authors,
I appreciate the efforts you have made to improve the manuscript. However, in my point of view, there are many issues that were not perfectly addressed or justified. Please, check the points below.
Issue: The research problem, according to the authors, is the absence of previous systematic reviews regarding team sports, since all the available literature is focused on soccer (Ometto et al., 2018; Sarmento et al., 2018, for example). However, if we have a lot of available information regarding soccer, why did you include this modality in the current study? Considering the gap you have mentioned, it would be better to include only other team sports.
Answer:Dear reviewer, we thank you for your suggestion, which we consider interesting and worthy of being taken into account for future research. However, we have considered including soccer studies because our intention is to give a more complete vision of the aspects that can be worked through the reduced games and to analyze its incidence in all the sports of a same category. Otherwise, since there is little research on other sports, vision would be reduced and less transferable to similar sports.
Attention: if you consider soccer for your study, you cannot consider the lack of studies as a problem, because there are a lot of previous reviews on this topic. There is a contradiction here. Rearrange the whole introduction to clear explicit the problem.
Issue: Depending on the date, there are many missing articles that comprise the eligibility criteria (just as an example, since many articles are missing: Bredt et al., 2018; Lemes et al., 2019; Moreira et al., 2019; Sousa, Bredt, Greco, Manuel Clemente, & Praça, 2019).
If you consider the papers published by Israel Teoldo, Gibson Praça and Filipe Clemente, (please, check the Research Gate for more details) you have more than 50 non-selected studies that accomplish with the eligibility criteria.
Answer: The manuscript search was carried out in four data bases (Web of Science, Scopus, SportDiscus and Pubmed) in May and June 2018. Perhaps, these studies had not been published or indexed yet.
We have not searched Research Gate, but we will consider it in future investigations. Thank you for your recommendation.
Attention: The search has been conducted more than 8 months ago. I don't think there is a strong rationale for this. In my opinion, the search must be redone considering the articles published until the date of the submission of the manuscript. Without this, we are going to have a recently published review with a non-up-to-date literature.
Issue: Mathematically talking, if the number of passes per minute per player in a 3v3 game is 3, and this number is 1 in a 9vs9 game if both games have the same time, the total number of passes registered in both games is equal (=18). So, the conclusion “the majority of studies agree that reducing the number of players leads to an increase of technical actions” is only true when talking about the relative number, no the absolute one (and you did not relativize the numbers in the current review study).
Answer:We have relativized the numbers about this issue.
Attention: this relativisation was not presented. You have only mentioned it (without explaining how this was done), which is not enough.
Issue: the absence of discussion.
Answer:We have looked for possible explanations to the results found.
Attention: only minor changes were made, not enough to provide a full discussion. The current discussion is, again, focused on representing the results instead of looking for explanations and comparing the data with previous revisions.
Author Response
Dear reviewer,
we have resubmitted the revised version of the manuscript. We have left in green the changes made in the first round, and in yellow those of the second round of review.
Thanks,
COVER LETTER
Manuscript ID: ijerph-714510. Type of manuscript: Article. Title: Small-sided games as a methodological resource for team sports teaching: A systematic review
|
Reviewer 1’s comments and suggestions for authors |
Details of the revisions and responses |
|
Line 14: replace work by stimuli. |
|
|
Attention: if you consider soccer for your study, you cannot consider the lack of studies as a problem, because there are a lot of previous reviews on this topic. There is a contradiction here. Rearrange the whole introduction to clear explicit the problem. |
We have clarified the rationale for the need for this systematic review.
|
|
Attention: The search has been conducted more than 8 months ago. I don't think there is a strong rationale for this. In my opinion, the search must be redone considering the articles published until the date of the submission of the manuscript. Without this, we are going to have a recently published review with a non-up-to-date literature. |
The search has been redone considering the articles published until the date of the submission of the manuscript. The new search incorporates 9 new articles from the years 2018 and 2019. |
|
Attention: this relativisation was not presented. You have only mentioned it (without explaining how this was done), which is not enough. |
We have explained how this relativisation was done.
“In this respect, it is important to note that this was the frequency of technical elements per player”. |
|
Attention: only minor changes were made, not enough to provide a full discussion. The current discussion is, again, focused on representing the results instead of looking for explanations and comparing the data with previous revisions. |
Now in the discussion we looked for explanations and compare the data with previous studies.
New references have been added in this regard. |
THANK YOU for your comments and suggestions.
